# Mediating Role of Job Satisfaction in the Relationship Between Leisure-Time Physical Activity and Emotional State of Healthcare Workers: A Cross-Sectional Survey

**DOI:** 10.3390/healthcare12232406

**Published:** 2024-11-29

**Authors:** Francesco Fischetti, Ilaria Pepe, Gianpiero Greco, Maurizio Ranieri, Luca Poli, Luigi Vimercati, Stefania Cataldi

**Affiliations:** 1Department of Translational Biomedicine and Neuroscience (DiBraiN), University of Study of Bari, 70124 Bari, Italy; francesco.fischetti@uniba.it (F.F.); ilaria.pepe@uniba.it (I.P.); maurizio.ranieri@uniba.it (M.R.); 2Section of Occupational Medicine, Interdisciplinary Department of Medicine, University of Study of Bari, 70124 Bari, Italy; luigi.vimercati@uniba.it; 3Department of Wellbeing, Nutrition and Sport, Pegaso Telematic University, 80143 Naples, Italy; stefania.cataldi@unipegaso.it

**Keywords:** work satisfaction, affective state, healthcare employees, emotional well-being, workplace stress

## Abstract

Background: Work-related stress among healthcare employees can lead to burnout, worsened mood, and job dissatisfaction. Although physical activity is known to enhance mood and mental health, its impact on job satisfaction and emotional well-being in healthcare workers is under-researched. This study aimed to explore the associations between leisure-time physical activity (LTPA), job satisfaction, and emotional state and to investigate the mediating role of job satisfaction in the effect of LTPA on the emotional state of healthcare workers. Methods: A self-administered questionnaire, including items on LTPA, job satisfaction, and the emotional state, was distributed to 98 healthcare workers affiliated with the Bari Polyclinic Hospital (Mean age = 46.3; SD = 15.4 years). Composite measures of global job satisfaction and emotional state were extracted by Factor Analysis using the principal components method. The relationship between LTPA, job satisfaction, and the emotional state was investigated through General Linear Model (GLM) mediation models. Results: There was no significant direct effect of LTPA on negative emotional states (β = −0.08, *p* = 0.37). However, job satisfaction significantly mediated this relationship (β = −0.09, *p* = 0.04), indicating that engaging in LTPA increased job satisfaction, which in turn reduced negative emotional states. Similarly, LTPA did not have a significant direct effect on positive emotional states (β = 0.06, *p* = 0.48), but it indirectly resulted in increased positive emotional states through its positive impact on job satisfaction (β = 0.12, *p* = 0.03). Conclusions: The findings indicate that LTPA indirectly influences both negative and positive emotional states through job satisfaction. Engaging in LTPA enhances job satisfaction, which subsequently leads to reductions in negative emotional states and increases in positive emotional states among healthcare workers. These results underscore the importance of promoting physical activity as a strategy to improve job satisfaction (JS) and the emotional well-being of healthcare workers.

## 1. Introduction

Healthcare workers in clinical settings, including physicians, physiotherapists, paramedics, nurses, and nursing assistants, experience significant job stress and mental exhaustion due to the demands of patient safety and consumer expectations [1,2,3]. These occupational stressors stem from various factors such as handling clinical obligations, excessive workloads, emotional strain, high patient volumes, long working hours, limited resources, and extended patient recovery times [1,3]. Consequently, workplace stress can lead to burnout and fatigue, both physically and emotionally [4,5].

Engaging in physical activity (PA) is a recognized coping strategy that may mitigate the impact of stress and burnout [6]. There is a strong correlation between higher levels of physical activity and better well-being, as well as a lower perception of stress at work [7,8]. Regular PA has been shown to positively impact lifestyle-related health issues such as obesity and cardiovascular disease [9,10]. Exercise and physical activity are consistently associated with improved mental health outcomes, including a lower risk of depression, better mood, and positive affect in both the general population and individuals with physical health issues [11,12,13]. Studies on various forms of exercise, such as aerobic exercise, resistance training, recreational sports, walking, and high-intensity interval training, along with observational studies comparing health outcomes among people with different levels of physical activity, have provided substantial evidence of the health benefits of exercise [14,15,16].

Despite the recognized benefits, research indicates that while some healthcare workers are physically active during their job duties, a significant portion engage in less physical activity and exercise during their free time than recommended [17]. This inactivity places healthcare workers at risk for physical inactivity-related issues. Engaging in leisure-time physical activity (LTPA) can enhance and promote a pleasant emotional state [18]. LTPA encompasses organized programs, involvement in sports, and lifestyle hobbies [19,20].

Previous studies have demonstrated the beneficial effects of LTPA on subjective well-being including levels of pleasure and life satisfaction [21,22,23]. While the impact of LTPA on health and leisure satisfaction has been extensively researched, less attention has been given to its role in job satisfaction. Job satisfaction, according to [24], is a pleasant or upbeat emotional state brought on by an evaluation of one’s work or work experiences. Previous studies have shown that job satisfaction significantly affects an individual’s productivity at work, as evidenced by increased job performance and decreased employee absenteeism [25,26,27]. Furthermore, job satisfaction is closely linked to improved mental and emotional health [6,28,29]. According to studies, there is a significant positive relationship between job satisfaction and life satisfaction [30,31,32], as well as psychological, emotional, and social well-being [33,34]. Employees who are satisfied with their jobs have been shown to carry these feelings outside of the work context [34]. Thus, understanding the factors that influence job satisfaction and strategies to enhance it is of great interest to both employers and employees.

The positive effects of LTPA on psychosocial outcomes have been well documented. Previous studies have shown that exercise can reduce work stress, improve inner calm and mood [35], positively influence self-concept [36], and enhance both individual and social resources [37,38]. Although the impact of physical activity on domains such as health and leisure satisfaction has been extensively studied, research on the relationship between physical activity and job satisfaction is relatively limited despite job satisfaction being a key indicator of overall life satisfaction.

Higher levels of LTPA may enhance job satisfaction through improved psycho-emotional well-being [12]. Additionally, studies have shown that emotional fatigue and stress after work are poor indicators of job satisfaction [39]. Engaging in LTPA can help employees manage and recover from stress by replenishing stress-related resources after work [6,40].

While existing research has demonstrated positive associations between leisure-time physical activity and emotional well-being, leisure-time physical activity and increased job satisfaction, and job satisfaction and a positive emotional state, no studies, to our knowledge, have examined the mediating effect of job satisfaction on the relationship between leisure-time physical activity and emotional well-being. This study aimed to fill that gap by exploring the mediating role of job satisfaction in the relationship between LTPA and emotional state among healthcare workers. By investigating this relationship, we seek to provide insights into how LTPA can be leveraged for the emotional well-being of healthcare workers through increasing job satisfaction as a mediator of the interaction. Based on the literature reviewed, we hypothesized that engaging in LTPA positively influenced the emotional state of healthcare professionals mediated by job satisfaction.

## 2. Materials and Methods

The reporting of this study follows the Strengthening the Reporting of Observational Studies in Epidemiology (STROBE) statement (Appendix A).

### 2.1. Study Design and Setting

This study was a cross-sectional observational analysis conducted as part of the ongoing project “Horizon Seeds S70” at the University of Bari, focusing on the “Well-being of Health Workers and Exercise”. Study participants were recruited from the University Hospital of Bari, Italy, specifically targeting healthcare workers across various departments. To ensure comprehensive participation, we launched a recruitment campaign throughout the hospital, utilizing extensive internal publicity.

### 2.2. Participants

Ninety-eight participants were interviewed. The mean age of the respondents was 46.3 years (SD = 15.39). Of the participants, 51 (52%) were women and 47 (48%) were men. Most healthcare workers were either single (40 or 40.8%) or married (36 or 37.7%). Professionally, most of the respondents (57.1%) were doctors. About half of the interviewees (47%) had a permanent contract, with an average of 10 years of service (SD = 11), and 53% worked in shifts, predominantly covering both day and night shifts (71%) (Table 1).

Participants were selected through a hospital-wide campaign. This campaign utilized internal communication channels, including personalized email notifications to all staff, bulletin board postings in common areas, and verbal announcements during departmental meetings. Healthcare workers interested in participating contacted the experimenter via email. Upon contact, they were provided with a link to the online questionnaire. Participants were selected if they met predefined eligibility criteria. This approach aimed to maximize reach and ensure a diverse representation of healthcare workers from different units within the hospital. The sampling method used was convenience sampling, where participants were selected based on their availability and willingness to participate during the study period. This non-probability sampling technique was chosen due to the practical constraints of engaging healthcare workers within a limited timeframe and the study’s focus on voluntary participation. While this approach limits generalizability, it is suitable for exploratory research aiming to understand associations within a specific context, such as a single hospital.

The study population consisted of 2.536 healthcare workers at the Bari Polyclinic University Hospital. To assess the representativeness of the sample, the semi-amplitude of the confidence interval was calculated a posteriori using the inverse formula for sample size determination. For the sample size of 98 participants, this resulted in a semi-amplitude of 10%, ensuring an acceptable level of reliability.

Regarding the eligibility criteria, healthcare workers were eligible to participate in this study if all the following criteria were met: (1) they were 18 years or older, (2) they worked in inpatient or outpatient healthcare facilities, (3) they had contracts or worked full-time during the study period, and (4) they had spent at least 3 months in their current healthcare unit.

### 2.3. Procedures

Data were collected between January and February 2024 using an online self-administered questionnaire distributed via Google Forms. Healthcare professionals were invited to participate through a hospital-wide recruitment campaign, and eligible participants completed the survey at their convenience. The questionnaire collected data on socio-demographic characteristics, job satisfaction, emotional state, and engagement in leisure-time physical activity (LTPA), which was self-reported.

All completed surveys were securely stored in a Google database, and the data were exported to an Excel spreadsheet (Microsoft Office 365; Microsoft Corporation, Redmond, WA, USA) for analysis. The procedures followed were in accordance with the ethical standards of the Helsinki Declaration and approved by the Ethics Committee of Bari University (protocol code 0030611|28/03/23).

### 2.4. Data Sources/Measurement

We created the questionnaire using Google Forms and distributed it to healthcare employees across various departments of Bari Polyclinic University Hospital. All participants were informed about the study and provided their consent for the processing of their data. The questionnaire, which was self-administered in Italian, was completed online.

A multi-stage process was employed to develop the questionnaire, ensuring its relevance and validity for the study’s objectives and healthcare context. Initially, a comprehensive literature review was conducted to identify key factors influencing job satisfaction and emotional states in healthcare workers. The theoretical framework was grounded in Herzberg’s two-factor theory [41,42], which distinguishes between hygiene factors (e.g., working conditions and interpersonal relationships) and motivators (e.g., achievements and recognition) as critical influences on job satisfaction. For instance, items such as “I am satisfied with my working environment” reflect hygiene factors that address conditions preventing dissatisfaction, whereas items like “I enjoy going to work” and “I have achievable goals” capture motivators that enhance satisfaction through personal growth and fulfillment. Herzberg’s theory remains a robust foundation for research on job satisfaction in various international contexts, particularly in healthcare settings, and has been widely applied in studies with nursing populations [42,43,44].

For emotional states, the Positive and Negative Affect Schedule (PANAS) [45] was used as a guiding framework. PANAS is a multidimensional measure that encompasses both positive (e.g., energy, enthusiasm) and negative (e.g., distress, agitation) emotional components, making it well suited for assessing affective states in professional environments.

Based on this theoretical foundation, a preliminary set of items was drafted to reflect the identified constructs of job satisfaction and emotional state. This draft underwent review by a panel of ten experts, including two healthcare professionals, two human resource psychologists, two psychotherapists, two statisticians with expertise in survey design, and two kinesiologists. The panel evaluated the relevance, clarity, and alignment of the items with the study’s objectives and the specific healthcare context.

The final questionnaire, administered online in Italian via Google Forms, consisted of three sections.

The first part gathered general and socio-demographic information, including age (years), sex (male or female), marital status (single, in a relationship, separated/divorced, married, and widowed), number of children, type of contract (fixed-term, permanent, collaboration, other), job role (nurse, doctor, healthcare assistant, other), length of time in the current job (years), and details about work shifts (yes/no) and their timings (day shifts/day-night shifts). Information on whether participants engaged in physical activity during their free time (e.g., “Do you engage in physical activity outside of work hours?”) (e.g., walking, jogging, cycling, sports, etc.), the amount of time dedicated to such activities in a week (<75 min, between 75 and 150 min, between 150 and 300 min, >300 min), and the frequency of days per week dedicated to LTPA was also collected. The second part of the questionnaire included eight questions that explored different domains related to job satisfaction among the sample. The third and final part consisted of seven items that assessed the emotional states experienced in the past four weeks (four items related to negative emotional states and three to positive emotional states).

### 2.5. Variables

The primary outcomes of interest were the emotional state of healthcare workers, measured through self-reported items in the questionnaire. The emotional state was assessed using a Likert scale ranging from 1 (never) to 5 (always), encompassing both positive and negative emotional experiences over the past four weeks. The predictor variable was LTPA, self-reported as a binary variable (yes/no) and converted into Bernoulli data (0 or 1), where 0 indicated not performing LTPA and 1 indicated performing LTPA. Job satisfaction was mediator variable. Responses were recorded on a Likert scale from 1 (never) to 5 (always), creating a composite measure of global job satisfaction through factor analysis.

Potential confounders included socio-demographic factors such as age, sex, marital status, job role, and work shift patterns. These variables were collected to control for their potential influence on the relationship between LTPA, job satisfaction, and emotional state.

### 2.6. Study Size

The necessary sample size was predetermined using G*Power version 3.1.9.7 [46]. The analysis was based on multiple linear regression using the two-tailed independent *t*-test, 95% confidence level, 5% margin of error and 95.0% power, and mean effect size of 0.15. The result indicated a required sample of 74 participants.

### 2.7. Statistical Analysis

Statistical analysis was performed using the IBM SPSS Statistics 25.0 program (IBM, Chicago, IL, USA). Quantitative data were represented by the mean and standard deviation whereas qualitative variables were summarized using absolute frequencies and percentages. The level of statistical significance was set a priori at *p* ≤ 0. 05.

In order to assess any potential effect of confounding variables (age, sex, marital status, type of contract, type of work shift) on the outcome, the Mann–Whitney U-test was used for two independent samples, the Kruskal–Wallis test for more than two independent samples, and the Spearman correlation coefficient for quantitative variables after determining the absence of normality of the two components of emotional state according to the Kolmogorov–Smirnov test.

To assess the validity and reliability of the job satisfaction and emotional state sections of the questionnaire, internal consistency and factorial validity were evaluated using Cronbach’s alpha and exploratory factor analysis (EFA), respectively.

Exploratory factor analysis (EFA) was applied to determine the validity of the questionnaire dimensions. Three preliminary steps were performed: (i) the Kaiser–Meyer–Olkin (KMO) sampling adequacy test, (ii) Bartlett’s test of sphericity, and (iii) the assessment of communalities. The KMO test was conducted for individual items and deemed adequate if the value was greater than 0.6 [47,48]. Bartlett’s test of sphericity was used to measure the overall significance of the correlations between all the elements of the measurement instrument. A significant *p*-value of less than 0.05 indicates that the data do not produce an identity matrix, implying a multivariate normal distribution suitable for EFA [49]. Communality refers to the proportion of common variance within an observed variable, and a communality value less than 0.3 indicates that the item does not align well with other items in its factor [48]. The number of factors to be retained in the model was determined using eigenvalues and the unweighted least squares method, alongside the scree plot and Kaiser’s rule, which extracts only factors with an eigenvalue of 1 or greater. Generally, an item loading greater than 0.3 is considered acceptable [50]. However, given the importance of item cross-loading, items with loadings closer to one are more significant, so we set a threshold for item loading at 0.5 or higher. Finally, the oblique Promax rotation was applied [51], and the related chi-square (χ^2^) and *p*-values were determined. Additionally, to measure the common variance not associated with the factors, the uniqueness index (1-communality) was calculated. The reliability or internal consistency of the completed questionnaire was assessed following the EFA. Internal consistency was measured using Cronbach’s alpha coefficient and tested for all items in the questionnaire. A Cronbach’s alpha value greater than 0.7 was considered acceptable while a value greater than 0.90 was deemed excellent [52].

A principal component analysis (PCA) was conducted on all 7 items related to emotional states. Both EFA and PCA identified two factors, corresponding to the two overarching dimensions of emotional states, with substantial loadings of each indicator on their respective constructs: positive and negative emotional states.

Similarly, PCA on all 7 items measuring job satisfaction revealed a factor structure consistent with the hypothesized construct. The loadings of each indicator on its respective construct were statistically significant. Overall, the exploratory factor analysis confirmed that each item was aligned with its hypothesized construct.

After conducting PCA on the items related to job satisfaction and perceived emotional state to extract a single “general factor” as a measure of job satisfaction and emotional state, these factor scores were utilized as both a mediator and a dependent variable in the mediation study. This study aimed to examine the possible indirect impact of job satisfaction on the relationship between LTPA and emotional state.

Since none of the covariates were statistically associated with each outcome (positive/negative emotional state), except for sex, this variable was introduced as a factor in both General Linear Model (GLM) mediation models.

In the mediation model, the global positive and negative emotional state components served as the dependent variable, the job satisfaction factor was the mediating variable, and LTPA was included as an independent dummy variable, as well as sex. The analysis reported the following effects—(i) total effects: these encompassed both direct and mediated effects; (ii) indirect effects: these highlighted the mediating role of job satisfaction; and (iii) direct effects: these indicated the relationship between LTPA and composite emotional state scores, and the relationship between sex and outcome, excluding the influence of job satisfaction. The mediation effect of job satisfaction was computed using the products-of-coefficients approach. Significance of mediation requires that its 95% confidence interval does not include 0.

## 3. Results

### 3.1. Sample Characteristics

As indicated in Table 1, a total of 98 healthcare workers were interviewed. The results of the Mann–Whitney and Kruskal–Wallis U-tests, as well as the Spearman correlations between potential covariates and the outcome, are presented in Table 1. In summary, no statistically significant associations were found between age, marital status, job role, type of shift, job title, and emotional state. However, a statistically significant association was identified with sex (U = 720; *p* < 0.001; U = 836, *p* = 0.010).

### 3.2. Questionnaire Validation

Table 2 shows the validity and reliability analysis results. As a result of the exploratory factor analysis for a validity check of the eight items in the job satisfaction section, each factor loading was above 0.60, confirming that there were no issues with validity after a Promax oblique rotation.

Each item was labeled as JS#, indicating aspects related to job satisfaction and the order of the items. The items were as follows:

JS1: “I am satisfied with my job position”.

JS2: “I like to go to work”.

JS3: “I have achievable goals”.

JS4: “I am satisfied with my working environment”.

JS5: “I have freedom to choose how to do my job”.

JS6: “I work at a sustainable pace”.

JS7: “I have awareness and clarity of my tasks and responsibilities”.

JS8: “I feel part of a team”.

The Kaiser–Meyer–Olkin (KMO) measure of sampling adequacy was reported at 0.86, ranging from 0.85 to 0.97, indicating that the items had been adequately sampled. Bartlett’s test of sphericity yielded χ^2^ (28.000) = 294.032 with *p* < 0.000, further supporting the validity of the factor analysis. As illustrated by the eigenvalue scree plot and following Kaiser’s rule, the presence of a single factor was revealed, which was subsequently labeled “global job satisfaction” (all item loadings ≥ 0.5).

Cronbach’s alpha value was used to analyze the reliability of the items, and the Cronbach’s alphas were all over 0.80, confirming that there was no problem with the reliability of this section (Table 2).

Likewise, following an oblique rotation of the Promax, each factor loading was greater than 0.60 according to the findings of the exploratory factor analysis for the validity check of the seven items linked to the study of the perceived emotional state, indicating that no validity issues existed.

Each item was labeled ES#, indicating aspects related to the participants’ emotional states perceived in the last 4 weeks. The items were as follows:

ES1: “He felt down in the dumps”.

ES2: “He felt agitated”.

ES3: “He felt exhausted”.

ES4: “He felt discouraged and sad”.

ES5: “He felt calm and peaceful”.

ES6: “He felt full of energy”.

ES7: “He felt lively and bright”.

Items were sufficiently sampled, as shown by the Kaiser–Meyer–Olkin (KMO) measure of sampling adequacy, which was reported at 0.85 (Table 2). Further confirming validity, Bartlett’s test of sphericity produced a result of χ^2^ (21.000) = 351.191 with *p* < 0.000. After applying the Promax rotation method, two factors (eigenvalues greater than 1 and all items loaded ≥ 0.5) satisfied our criteria. We also assessed Cronbach’s alpha on seven items. The internal consistency was calculated to be 0.88, which suggested that the items were reliable indicators of the emotional state assessed.

### 3.3. Relationship Between LTPA, Job Satisfaction, and Emotional State

A single explanatory component of job satisfaction and two components characterizing emotional state were extracted using PCA with Varimax rotation. This was supported by quantitative indices such as the scree plot and eigenvalues greater than one. Additionally, there was a theoretical need for a single score to represent job satisfaction and two scores to represent the two perceived emotional states.

For the work context items (Table 3), the single component job satisfaction (eigenvalue = 4.13) explained 52% of the variance. For items related to emotional states, Table 3 suggests that three items were included in factor 2 (positive emotional state) and four items were included in factor 1 (negative emotional state).The first component (eigenvalue = 4.12) accounted for 40% of the variance and showed a positive correlation with all items investigating negative emotional states; the second component (eigenvalue = 1.2) accounted for 33% of the variance and showed a positive correlation with items investigating positive emotional states. The component loadings are displayed in Table 3.

In the first GLM mediation analysis, there was no significant direct effect of LTPA on negative emotional states (β = −0.077, *p* = 0.367) (Table 4). However, job satisfaction significantly mediated this relationship. Specifically, the significant overall negative indirect effect indicated that engaging in physical activity during leisure time increased job satisfaction (β = 0.217, *p* = 0.02). In turn, higher job satisfaction was associated with reduced negative emotional states (β = −0.459, *p* < 0.001). Therefore, indirectly, practicing physical activity during leisure time led to reductions in negative emotional states through its positive impact on job satisfaction (β = −0.099, *p* = 0.04) (Table 4).

The second GLM mediation model, run in parallel, revealed that job satisfaction positively influenced the relationship between practicing physical activity in leisure time and an increase in the perceived positive emotional state (Table 5). Although LTPA did not have a significant direct effect on positive emotional states (β = 0.0585, *p* = 0.479), it indirectly resulted in increased positive emotional states through its positive effect on job satisfaction (β = 0.120, *p* = 0.03). (Table 5).

For negative emotional states, sex had a significant direct effect (β = −0.250, *p* = 0.003), but job satisfaction did not mediate this effect (β = −0.0625, *p* = 0.178). The detailed results are presented in Table 4 and Table 5.

## 4. Discussion

### 4.1. Key Results

This study aimed to explore the mediating role of job satisfaction in the relationship between LTPA and the emotional state of healthcare workers; the main result was that while LTPA did not directly impact on the emotional states of healthcare workers, it had a significant indirect effect through job satisfaction. Within this analysis, we found that while LTPA did not have a direct effect on negative emotional states, it significantly influenced job satisfaction. Increased job satisfaction, in turn, was associated with lower levels of negative emotional states. This suggests that the positive impact of LTPA on emotional well-being is mediated by its ability to enhance job satisfaction. By engaging in physical activity during their leisure time, healthcare workers can boost their job satisfaction, which subsequently leads to a reduction in negative emotional states. Similarly, LTPA did not directly influence positive emotional states. However, job satisfaction again played a crucial mediating role. Higher job satisfaction from LTPA was associated with increases in positive emotional states. Sex significantly affects emotional states. The effects of sex on emotional states are direct and not mediated by job satisfaction, highlighting the importance of considering sex differences when developing interventions to improve emotional well-being through physical activity.

### 4.2. Strength and Limitations

The current study, like other studies, was not without weaknesses. Firstly, the cross-sectional design of the study made it more difficult to determine the relationship among emotional states, job satisfaction, and LTPA. The causal directions of these associations must be confirmed by longitudinal or experimental methods, even if mediation analysis might offer insights into possible paths. Second, while 98 healthcare workers constitute a sufficient sample size for the statistical analyses conducted, the validity of the study and the statistical power were potentially compromised by a small sample size due to a lower response rate from participants, which is common in online surveys. To confirm these findings, future studies with larger samples are recommended. The sample may not have accurately represented the experiences of healthcare professionals in other areas or in diverse healthcare settings because it was collected from a single hospital in Bari, Italy. This limitation affected the external validity of the study. Finally, the study relied on a non-probabilistic convenience sampling method, which may have limited the representativeness of the sample.

### 4.3. Interpretation

Our study confirmed the hypothesis that LTPA positively impacts emotional state indirectly through job satisfaction. Participation in LTPA was linked to enhanced job satisfaction, which subsequently led to improved emotional states.

Despite the well-documented benefits of physical activity on mental health, our findings revealed no significant direct effect of LTPA on emotional states. Healthcare workers frequently work in high-stress conditions, which could reduce the immediate emotional benefits of physical activity. The chronic stress associated with demanding job duties may outweigh the short-term emotional temporary relief afforded by LTPA. These findings support the view that employees with low control and high requirements may have an entirely separate connection between activity and health outcomes. Ref. [53] found that nurses working in shifts experienced less mental fatigue when they perceived greater rewards and had more control over their work. This reduction in mental fatigue was not related to the amount of physical energy they expended.

Healthcare workers with high levels of baseline stress may need more significant or frequent physical activity to experience noticeable emotional benefits. Consequently, cumulative stress might diminish the immediate impact of LTPA on emotional states [54].

Healthcare professionals who directly care for patients are likely to experience certain occupational features in the workplace like mental and physical pressures. The relationships between leisure-time physical activity and employee outcomes in the sample may be influenced by these demand and control characteristics identified in employment roles and responsibilities. Furthermore, it has been demonstrated that occupational physical activity mitigates the impact of leisure physical activity, implying that when employees engage in high levels of occupational physical activity, the correlations between leisure physical activity and emotional states decrease [55,56].

Lastly, there are additional aspects of the workplace that could have an impact on employees’ psychological well-being. Interactions with coworkers, individual motivation, job tasks and responsibilities, workplace culture and practices, autonomy, medical system hierarchy, and perceived and real social support in the workplace are a few examples of potential contributing factors [57,58].

According to [59], engaging in physical activity within a socially supportive atmosphere has been shown to yield more psychological advantages than engaging in such activity alone. It is possible that the immediate benefits on emotional state of LTPA will be less noticeable if healthcare professionals participate in it in an environment devoid of support or without enough social interaction.

Based on the organizational setup and work climate, our results showed that LTPA positively influences emotional states indirectly through its impact on job satisfaction. Engaging in LTPA was associated with increased job satisfaction, which in turn improved emotional states. This mediated relationship highlights the critical role that job satisfaction plays in the well-being of healthcare workers.

The physical benefits of regular exercise, such as increased energy levels and reduced fatigue, can enhance job performance and productivity. This increased productivity and the sense of accomplishment associated with completing work tasks more effectively can directly boost job satisfaction [60,61].

Engaging in physical activity during leisure time provides employees with a sense of control and autonomy as they can choose their preferred activities and set their own fitness goals. Autonomy and control are closely linked to greater job satisfaction as they help employees feel more responsible and less stressed [62,63]. This enhanced job satisfaction, in turn, positively impacts mental health by reducing stress and improving overall well-being and emotional states [29,31,34].

Self-determination theory emphasizes the importance of autonomy in promoting psychological health [64]. When employees engage in self-chosen physical activities, they meet their intrinsic need for autonomy, which can enhance their overall sense of well-being and job satisfaction [65]. Higher job satisfaction then acts as a buffer against stress and promotes mental health. This is supported by the conservation-of-resources theory [66], which posits that individuals strive to obtain, retain, and protect valuable resources, including job satisfaction. When job satisfaction is high, employees have more psychological resources to cope with stressors, thereby improving their mental health [67].

Additionally, the job demands–resources model [68] provides a framework for understanding how job satisfaction mediates the relationship between leisure-time physical activity and mental health. According to this model, job resources such as autonomy and job satisfaction help mitigate the negative effects of job demands on mental health [69]. Physical activity, by enhancing job satisfaction, acts as a job resource that buffers against work-related stress and promotes mental well-being [70].

Interestingly, our mediations models revealed that sex has a direct impact on emotional states, independent of job satisfaction. This suggests that while job satisfaction is a significant mediator for the relationship between LTPA and emotional states, sex plays a distinct and separate role in influencing emotional well-being. Sex differences in coping mechanisms contribute to the distinct emotional responses observed in healthcare workers [71]. Women are more likely to use emotion-focused coping strategies, such as seeking emotional support and reflecting on feelings, which can sometimes lead to greater emotional awareness but also heightened emotional distress [72,73]. In contrast, men may adopt problem-focused coping strategies, aiming to address the source of stress directly, which might reduce the immediate emotional impact but could lead to the under-recognition of emotional strain [74].

## 5. Conclusions

The indirect effect of LTPA on emotional well-being through job satisfaction was first investigated and confirmed in our study. Engaging in physical activity during leisure time enhances autonomy, competence, and positive emotions, leading to higher job satisfaction. This job satisfaction, acting as a psychological resource, helps buffer against job stress and emotional exhaustion, thereby improving overall emotional well-being. Our study underscores the importance of job satisfaction as a mediator in the relationship between LTPA and emotional states. While LTPA alone may not directly improve emotional states, its positive impact on job satisfaction can lead to significant improvements in emotional well-being. These findings suggest that interventions aimed at enhancing job satisfaction could amplify the emotional benefits of physical activity for healthcare workers. Future research should continue to explore these dynamics and develop strategies to promote both physical activity and job satisfaction to improve overall well-being in high-stress professions.

## Figures and Tables

**Table 1 healthcare-12-02406-t001:** Participant characteristics (n = 98).

	All Participants	Negative Emotional State	Positive Emotional State
Covariates	N (%)	*p* Value	*p* Value
Years of age (mean ± SD)	46.3 ± 15.4	0.39 *	0.10 *
Female sex (%)	52	<0.001 **	0.01 **
Single marital status (%)	40.8	0.90 ***	0.18 ***
Permanent contract (%)	46.9	0.39 ***	0.44 ***
Doctors (%)	57.1	0.07 ***	0.43 ***
Years in the current job (mean ± SD)	13.9 ± 12.9	0.46 *	0.07 *
Day and night shifts (%)	71.2	0.23 **	0.34 **

Notes: * Spearman correlation; ** Mann–Whitney statistics; *** Kruskal–Wallis’s statistics; SD, standard deviation.

**Table 2 healthcare-12-02406-t002:** Results of validity and reliability tests.

Items	Loadings	Eigen Value	The Kaiser–Meyer–Olkin (KMO) Value	Cronbach’s α
JS1	0.76		0.85	
JS2	0.70		0.84	
JS3	0.68		0.81	
JS4	0.68	4.127	0.84	0.86
JS5	0.68		0.85	
JS6	0.63		0.89	
JS7	0.61		8.60	
JS8	0.60		0.89	
ES1	0.90		0.82	
ES2	0.64		0.88	
ES3	0.69		0.83	
ES4	0.88	4.115	0.81	0.88
ES5	0.70		0.87	
ES6	0.76		0.86	
ES7	0.88	1.207	0.82	

**Table 3 healthcare-12-02406-t003:** Results of principal component analysis.

Items	Factor 1	Factor 2	Uniqueness
JS1	0.78		0.39
JS2	0.75		0.44
JS3	0.73		0.47
JS4	0.73		0.47
JS5	0.72		0.48
JS6	0.69		0.52
JS7	0.67		0.55
JS8	0.66		0.56
ES1	0.84		0.24
ES2	0.75		0.33
ES3	0.80		0.32
ES4	0.84		0.24
ES5		0.80	0.30
ES6		0.80	0.26
ES7		0.85	0.21

Notes: applied rotation method was Varimax.

**Table 4 healthcare-12-02406-t004:** Mediation analysis results.

					β	95% CI Lower	95% CI Upper	z-Value	*p*
Direct effects									
LTPA	→	Negative emotional state			−0.077	−0.544	0.2011	−0.90	0.367
Sex	→	Negative emotional state			−0.250	−0.827	−0.1703	−2.97	0.003
Indirect effects									
LTPA	→	Job satisfaction	→	Negative emotional state	−0.099	−0.434	−0.0095	−2.04	0.041
Sex	→	Job satisfaction	→	Negative emotional state	0.136	−0.111	0.6531	1.39	0.164
Component									
LTPA	→	Job satisfaction			0.217	0.055	0.9105	2.21	0.027
Job satisfaction	→	Negative emotional state			−0.459	−0.627	−0.2909	−5.34	<0.001
Sex	→	Negative emotional state			0.136	−0.111	0.6531	1.39	0.164
Total effects									
LTPA	→	Negative emotional state			−0.176	−0.808	0.0218	−1.85	0.063
Sex	→	Negative emotional state			−0.312	−0.994	−0.2518	−3.28	0.001

Notes: β, beta coefficient; CI, confidence interval; z-value, standardized statistic; *p*, *p*-value.

**Table 5 healthcare-12-02406-t005:** Mediation analysis results.

					β	95% CI Lower	95% CI Upper	z-Value	*p*
Direct effects									
LTPA	→	Positive emotional state			0.058	−0.230	0.49	0.707	0.479
Sex	→	Positive emotional state			0.144	−0.030	0.60	1.775	0.076
Indirect effects									
LTPA	→	Job satisfaction	→	Positive emotional state	0.120	0.018	0.51	2.103	0.035
Sex	→	Job satisfaction	→	Positive emotional state	0.075	−0.066	0.36	1.361	0.174
Component									
LTPA	→	Job satisfaction			0.217	0.055	0.9105	2.21	0.027
Job satisfaction	→	Positive emotional state			0.553	0.390	0.71	6.64	<0.001
Sex	→	Positive emotional state			0.136	−0.111	0.653	1.39	0.164
Total effects									
LTPA	→	Positive emotional state			0.178	−0.028	0.82	1.82	0.068
Sex	→	Positive emotional state			0.219	0.056	0.81	2.25	0.024

Notes: β, beta coefficient; CI, confidence interval; z-value, standardized statistic; *p*, *p*-value.

## Data Availability

The data presented in this study are available on request from the corresponding author. The data are not publicly available due to privacy.

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
