# Peer review of "Mediating Role of Job Satisfaction in the Relationship Between Leisure-Time Physical Activity and Emotional State of Healthcare Workers: A Cross-Sectional Survey"

_healthcare, 2024, doi:10.3390/healthcare12232406_

Round 1
Reviewer 1 Report
Comments and Suggestions for Authors
1. Title: please specify the cross-sectional survey rather than referring to it as “a questionnaire survey”.
2. Abbreviation: (1) Please provide the full name of “GLM Mediation Models” in abstract and main body.
(2) Please use the abbreviation of LTPA after providing the full spelling the first time.
3. Tables:
(1) Please develop your table layout based on the rules of statistical table/three-line table, particularly for Table1、Table2 and Table3.
(2) Table1: The terms “Outcome” and “emotional” appear to be redundant; “Mean ± SD” is only applied to the indicators “age” and “Length of time in the current job”, they are incorrect as column titles, and should be expressed as “Years of age (Mean ± SD)” and “Years in the current job(Mean ± SD)”.
(3) Please replace the column title “Mean ± SD” with “N(%)”. Consequently, the line title and figure should change accordingly, for example, “Marital status (% single)” and “40.8%” would become “Single marital status(%)” and “40 (48)”.
4. Limitation: Please describe the limitations of this study, for instance, the small sample size of 98 healthcare workers.
Reviewer 2 Report
Comments and Suggestions for Authors
Dear authors,
Congratulations on the article developed.
The article is well written overall, but needs minor rewording. I leave suggestions for improvement:
Title: it is unclear… with English errors and very long. It is suggested (validate the correct translation into English):
Mediating Role of Job Satisfaction in the Relationship between Leisure-Time Physical Activity and Emotional State of Healthcare Workers
Keywords: have little sensitivity to the variables under study. Mesh-Decs health sciences descriptors can be more specific and used: https://decs.bvsalud.org/en/ths/resource/?id=7756&filter=ths_termall&q=job%20satisfaction
Method:
It is reproducible and ethical assumptions are assured.
The sample is clear, but what is the population size? It must be clear!
Results: I did not identify errors in the statistical analysis and interpretation of the results.
Discussion: Overall, good.
Conclusions: Overall, good.
References:
The numbering of references appears in duplicate. To correct!
Recent and relevant references to the topic, but there are many.
Confirm that all references in the list are cited.
Check, throughout the text, whether the citations correspond correctly to the references.
Best wishes for the future!
Reviewer 3 Report
Comments and Suggestions for Authors
Title:
- Line 3: There is an error in “Leisure-TType.”
- Line 3: The title includes the term “Paperime,” which does not appear in the paper itself. Is this a mistake?
Abstract:
- Line 26: "GLM Mediation Models" could be clarified as "General Linear Model (GLM) mediation analysis" to specify the methodology more directly.
Introduction:
- Line 68: Consider replacing “numerous studies” with “previous research” or providing specific examples for a more precise introduction.
- Line 99: The objective stated in the abstract differs somewhat from the one in the introduction.
Methodology:
- Line 115: “Utilising extensive internal publicity” is unclear. Could you provide more specifics on how participants were recruited to the survey?
-
- Line 122: The use of the term “recruited” is confusing given the publicised “hospital-wide campaign utilising internal communication channels, including email notifications, bulletin board postings, and announcements at departmental meetings.” Could you clarify how the participants were selected as a convenience sample, and whether this sampling method is appropriate for the study?
-
- Lines 161-164: You mention developing two questionnaires, but it would be beneficial to explain the creation process for each. Which sources or frameworks did you use to develop the items, and what steps were taken to finalise the instruments? Were they reviewed by specialists to ensure appropriateness for the study’s context? This section needs further development.
- Additionally, it would be helpful to create a dedicated section for procedures within the materials and methods section. The procedural steps currently appear throughout the methodology section (e.g., Lines 115-119, Lines 131-135, Lines 142-144) and would be clearer if grouped into one cohesive section.
Results:
- Line 250: The section “Sample Characteristics of Participants” does not align with the study’s primary objectives. Consider moving this information to the Methodology under section 2.2 (Participants).
- Line 265: You discuss the validation of the questionnaire (first 8 items and then 7 items), but this was not listed as an objective of your study. Please clarify this section for consistency.
Discussion:
- Line 349: Sections 4.1 Key Results and 4.2 Strengths and Limitations would be better placed at the end of the discussion.
Citation and Referencing:
- Line 215: (Brown, 2009)
- There appears to be an error in the reference list with duplicate reference numbers.
Round 2
Reviewer 3 Report
Comments and Suggestions for Authors
Accepted the responses from the authors